# Anthocyanins Profiling Analysis and RNA-Seq Revealed the Dominating Pigments and Coloring Mechanism in Cyclamen Flowers

**DOI:** 10.3390/biology11121721

**Published:** 2022-11-28

**Authors:** Demei Xia, Guoqiang He, Kai Wang, Taoyuan Wang, Zhiguo Zhu, Zhaoqian Niu, Gongfa Shi, Guiling Liu

**Affiliations:** 1College of Landscape and Horticulture, Wuhu Institute of Technology, Wuhu 241003, China; 2Garden Center, Shenzhen University, Shenzhen 518060, China; 3College of Landscape Architecture, Northeast Forestry University, Harbin 150040, China

**Keywords:** anthocyanins profiling, RNA-Seq, *Cyclamen purpurascens*, anthocyanins-3-O-rutinoside, g8206_i0

## Abstract

**Simple Summary:**

Flower color is a complex plant trait that is mainly controlled by the accumulation of pigments, such as anthocyanins. However, the great diversity of flower color in plants cannot be attributed to a single pigment. Therefore, a more comprehensive approach is needed to fully understand the whole pigment spectrum. In this study, we used metabolomics to profile more than 100 plant pigment in red-flowered cyclamen. By comparing the anthocyanins metabolome in white-flowered cyclamen, we were able to identify key anthocyanins that are highly abundant in red flowers, but low in white flowers. Thus, we can pinpoint the pigments underlying specific flower color phenotype. To further reveal the gene expression network that ultimately controlling the production of pigments, we also performed RNA-Seq using the same plant materials. The transcriptomics revealed a significant altered gene expression profile between red flowers compared to white flowers. More importantly, differential expression analysis allowed us to identify key genes governing the pigment metabolome and flower color. Collectively, our study significantly advanced our understanding of the molecular basis of flower color in cyclamen.

**Abstract:**

Pigments in cyclamen (*Cyclamen purpurascens*) endows flowers with great ornamental and medicinal values. However, little is known about the biosynthetic pathways of pigments, especially anthocyanins, in cyclamen flowers. Herein, anthocyanins profiling and RNA-Seq were used to decipher the molecular events using cyclamen genotypes of red (HXK) or white (BXK) flowers. We found that red cyclamen petals are rich in cyanidin-3*-O*-rutinoside, cyanidin-3-*O*-glucoside, delphinidin-3-*O*-glucoside, malvidin-3-*O*-glucoside, peonidin-3-*O-*rutinoside, quercetin-3-*O*-glucoside, and ruti. In addition, our transcriptomics data revealed 3589 up-regulated genes and 2788 down-regulated genes comparing the BXK to HXK. Our rich dataset also identified eight putative key genes for anthocyanin synthesis, including four chalcone synthase (CHS, g13809_i0, g12097_i0, g18851_i0, g36714_i0), one chalcone isomerase (CHI, g26337_i0), two flavonoid 3-hydroxylase (F3′H, g14710_i0 and g15005_i0), and one anthocyanidin synthase (ANS, g18981_i0). Importantly, we found a 2.5 order of magnitude higher expression of anthocyanin 3-*O*-glucosyltransferase (g8206_i0), which encodes a key gene in glycosylation of anthocyanins, in HXK compared to BXK. Taken together, our multiomics approach demonstrated massive changes in gene regulatory networks and anthocyanin metabolism in controlling cyclamen flower color.

## 1. Introduction

Cyclamen (*Cyclamen persicum* Mill.) is a pot crop extensively cultivated worldwide [1,2,3]. The flower color spans from white to purple, reflecting great genetic diversity and creating great commercial value. The color variation stems mainly from pigment components that are derived from flavonoids, especially anthocyanins [1,2,3].

Flavonoids are water-soluble pigments synthesized from 4-coumaroyl-coenzyme A (CoA; via the phenylpropanoid metabolism) and malonyl-CoA (via the fatty acid synthesis pathway) through phenylprop a metabolism [4]. Six major subgroups of flavonoids exist in the leaf, flower, seeds, and fruit in higher plants: the colorless chalcones, flavones (celery), flavonols (tea, apple, wine), flavandiols, anthocyanins (wine, blueberry), and proanthocyanidins/condensed tannins. Besides, isoflavonoids had been found in legumes and some nonlegume plants, while 3-deoxyanthocyanins are present in *Sorghum bicolor*, *Zea mays*, and *Sinningia cardinalis* [5]. The biosynthetic pathway of flavonoids had been established by characterizing mutants deficient in the key enzymes in multiple species [6,7,8,9]. For anthocyanins, chalcone synthase (CHS) catalyzes the reaction between malonyl-coenzyme A (CoA, derived from the fatty acid synthesis pathway) and 4-coumaroyl-CoA (derived from the phenylpropanoid metabolism) to form naringenin chalcone [10]. Next, CHI (chalcone isomerase) catalyzes the transition from naringenin chalcone to produce naringenin flavanone [11]. Under the direction of F3H (flavanone-3-hydroxylase), F3′5′H (flavonoid-3′5′-hydroxylase) and F3′H (flavonoid-3′-hydroxylase), the naringenin flavanone is converted to dihydroflavonols [12]. DFR (dihydroflavonol 4-reductase) catalyzes the formation of leucoanthocyanins, while ANS/LDOX (anthocyanidin synthase/leucoanthocyanidin dioxygenase) covert leucoanthocyanins to anthocyanidins (delphinidin, cyanidin, pelargonidin) [8,13]. In addition, UFGT (uridine diphosphate-glucose:flavonoid 3-O-glucosyltransferase) catalyze anthocyanidins to produce glycosylated anthocyanins, such as delphinidin-glycosides, cyanidin-glycosides, and pelargonidin-glycosides [8]. The anthocyanins may be further methylated by OMTs (O-methyltransferases), leading to the formation of O-methylated anthocyanins (malvidin-glycosides, peonidin-glycosides, petudin-glycosides) [8,14].

Anthocyanins are the most important flavonoid pigments widely distributed in nature, dressing the flowers of plants with colors ranging from pink, red, magenta, purple, and blue to blue-black [15,16]. In addition to serve as coloring pigments for attracting dispersers and pollinators, anthocyanins also play a role in protecting photosynthetic tissues from light stress [17]. Moreover, anthocyanins benefit human health given their potential functions in anti-oxidation, anti-ageing, retinal protection, anti-cancer, and hypolipidemia [5,15]. Chemically, they are glycoside derivatives of 2-phenylbenzopyrylium or flavylium. More than thirty types of monomeric anthocyanidins or aglycone had been described with variations in the number of hydroxyl groups, position, number, and kinds of glucoside attached, and nature and number of aliphatic acids (or aromatic) attached to the glucoside [18,19]. Cyanidin, delphinidin, pelargonidin, peonidin, malvidin, and petunidin are six common types of anthocyanidins in various parts of plants.

The anthocyanin biosynthetic pathway is strictly regulated at the transcriptional level [8,14]. Transcription factors (TFs), including those in the MYB and bHLH families, have been shown to regulate the gene expression profiles of the pathway. By either positively or negatively controlling the gene expression level, they regulate anthocyanin biosynthesis and determines pigmentation patterns [8,14,20,21,22,23,24,25].

Anthocyanins, especially malvidin, peonidin, and cyanidin derivatives, are the predominant pigments in cyclamen flowers [1,2,3,4,5]. Herein, the anthocyanin metabolic profiling was performed in cyclamen of different colors (red or white) flowers to explore the dominating pigments. Moreover, comparative transcriptome analysis was performed to explore key enzymes participating in the synthesis of relevant pigments (anthocyanins).

## 2. Materials and Methods

### 2.1. Plant Materials and Growth Conditions

Two cyclamen (*Cyclamen persicum* Mill.) varieties with white (BXK) or red (HXK) colors were grown in the orchard of Nursery of Wuhu Vocational and Technical College, respectively. Flowers were snap frozen in liquid nitrogen immediately after collection and stored at −80 °C for until use. Three biological repetitions were performed for all experiments.

### 2.2. RNA Sequencing

RNA sequencing was performed by Wuhan MetWare Biotechnology. Total RNA was extracted using RNAprep Pure Plant Plus (DP441, TIANGEN, Beijing, China) and then enriched for total mRNA using poly-T oligo-attached magnetic beads. For library construction, NEBNext1 Ultra™ RNA Library Prep Kit (NEB, Ipswich, MA, USA) was used according to the kit instructions. Briefly, mRNA was fragmented and first strand cDNA was synthesized using random hexamer primer and M-MuLV Reverse Transcriptase, followed by second strand cDNA synthesis. cDNA fragments were then methylated at the 3′ ends and then ligated to adaptor for hybridization. Purificaiton was performed using AMPure XP beads (Beckman Coulter, Beverly, MA, USA). Then, PCR was performed after adding the Index (X) Primer, Universal PCR primers and High-Fidelity DNA polymerase. PCR products were purified and then assessed by Agilent Bioanalyzer 2100 system. Sequencing was performed on Illumina HiSeq2500™ (Illumina, San Diego, CA, USA). The entire experiment was repeated three times.

### 2.3. Splicing of Transcripts, Reads Mapping, Gene Expression Level Quantifying, and Differential Expression Analysis

Reads were filter by removing low quality ones (>10% unknown nucleotides or >50% low quality nucleotides). Then, transcripts were spliced using Trinity. The spliced transcripts were regarded as the reference sequences and the beads were mapped to the spliced transcripts using Hisat2. Relative gene expression was denoted with FPKM. Differential expression analysis was performed using DEGSeq. Two criteria were used for defining differentially expressed genes (DEGs): Fold Change ≥2 or ≤−2, and FDR < 0.01.

### 2.4. KEGG Pathway Enrichment Analysis

Kyoto Encyclopedia of Genes and Genomes (KEGG) analysis of differentially expressed genes was performed using http://www.genome.jp/kegg/ (accessed on 1 January 2022) and KOBAS software1.0 [26].

### 2.5. qRT-PCR Analysis

The expression of g1022_i0, g10365_i0, g10266_i0, g10340_i0, g10589_i0, g11358_i0, g34436_i0, g18620_i0, g526_i0, g5626_i0, and g8206_i0 was analyzed by real-time quantitative RT-PCR. cDNAs were synthesized using the iScript Reverse Transcription Supermix. qRT-PCR was performed on a Opticon 2 system (MJ Research, St. Bruno, QC, Canada). Each reaction was performed in triplicate, and eEF1α was chosen as the reference gene. The expression levels of the tested transcripts were calculated by the 2^−∆∆CT^ method [27]. All primers were designed using primer3.0 software (https://primer3.ut.ee/ (accessed on 1 January 2022) and listed in Appendix A.

### 2.6. Anthocyanin Metabolic Profiling Analysis Using LC-MS/MS

#### 2.6.1. Sample Extraction

In total, 100 mg powder was dissolved in 1.0 mL Buffer A (70% aqueous methanol). Samples were incubated overnight at 4 °C, and then centrifugated at 10,000× *g* for 10 min. The supernatant was filtrated using 0.22 μm pore size (SCAA-104, ANPEL, Shanghai, China). Then, HPLC-MS/MS analysis was performed LC-ESI-MS/MS consisting of Shim-pack UFLC SHIMADZU CBM30A coupled to 6500 Q TRAP.

#### 2.6.2. HPLC Conditions

HPLC was performed according to a published method [16]. The injection was 2 μL. The column was 10 cm long HSS T3 C18 (Waters, 1.8 µm, 2.1 mm. The gradient was: 100% of A (water, 0.04% acetic acid) at 0 min, 95% of B (acetonitrile, 0.04% acetic acid) at 11 min, 95% of B at 12 min, 5% of B at 12.1 min (till 15 min). The flow rate was 0.40 mL/min and temperature was 40 °C.

#### 2.6.3. Mass Spectrometry

Mass spectrometry was performed on 6500 QTRAP equipped with an ESI Turbo Ion-Spray interface as described. Positive ion mode was used. Analyst 1.6.3 software (AB Sciex, Framingham, MA, USA) was used for setting up instrument method and data acquisition [23].

#### 2.6.4. Metabolite Identification and Quantitative Analysis

Metabolites identification and quantification were performed using MultiaQuant software 3.0.3 with the MWDB database (Metware Biotechnology, Woburn, MA, USA). The spectra of each metabolite were calibrated based on peak pattern and retention time. The reproducibility among different samples was determined by overlapping display the total ion chromatography (TIC) [28,29,30,31].

## 3. Results

### 3.1. LC-MS/MS Analyzes the Anthocyanin Metabolic Profiling in Two Cyclamen Varieties with Different Colors

In this study, metabolic analysis of anthocyanin was carried out using an LC-MS-based metabolomics approach.

As shown in Appendix A, 108 anthocyanin metabolites, including 17 Cyanidin metabolites, 16 Delphinidin metabolites, 13 Malvidin metabolites, 19 Pelargonidin metabolites, 28 Peonidin metabolites, 6 Procyanidin metabolites, and 9 other flavonoid metabolites were detected and quantified. Hierarchical clustering analysis showed a consistent pattern of metabolite profiles among biological repetitions within each variety (Figure 1, Appendix A).

Orthogonal partial least squares discriminant analysis (OPLS-DA), which could maximize the difference among groups, was employed to screen differential anthocyanin metabolites. As shown in Table 1, 45 significantly different anthocyanin metabolites (including 37 up-regulated and 8 down-regulated) were identified. As expected, the content of most anthocyanin metabolites in flowers of “HXK” is higher than that in “BXK”, especially for cyanidin-3-*O*-rutinoside, cyanidin-3-*O*-glucoside, delphinidin-3-*O*-glucoside, malvidin-3-*O*-glucoside, peonidin-3-*O*-rutinoside, quercetin-3-*O*-glucoside, and rutin (Table 1). The content of these 3-*O*-glucoside or 3-*O*-rutinoside glycosylated cyanidin, delphinidin, malvidin, pelargonidin, and peonidin is prominent higher than the derivatives of them, suggesting that they may be the main reason for the color difference between red and white flowers. Among them, cyanidin-3-*O*-rutinoside, cyanidin-3-*O*-glucoside, malvidin-3-*O*-glucoside, and peonidin-3-*O*-rutinoside, which contribute to the red color, are the most abundant anthocyanin in red flowers.

### 3.2. RNA Sequencing Analysis of Flowers of Two Cyclamen Varieties with Different Color

Six cDNA libraries derived from flowers of HXK and BXK (namely HXK_1, HXK_2, HXK_3, BXK_1, BXK_2, and BXK_3, respectively) were constructed and sequenced. After quality control, 14,8922 transcripts, 11,0942 unigenes, and 21,622 conden genes were acquired (Table 2). The detailed sequence of spliced transcripts, unigenes, and conden genes are presented in Dataset S2, S3, and S4. Detailed information of beads and probability of mapping of the six samples are presented in Table 3.

DEGs, including 3589 up-regulated genes and 2788 down-regulated genes, were detected comparing the two varieties (Figure 2A, Dataset S5). To verify the RNA-Seq data, the expression level of nine randomly chosen DEGs was validated. As shown in Figure 2C, qRT-PCR analysis showed that the relative expression patterns of the genes were consistent with RNA-Seq data.

### 3.3. KEGG Analysis of DEGs

Functional subcategories of DEGs were shown in Figure 2B. The most enriched 24 KO terms were beta-alanine metabolism, stilbenoid, diarylheptanoid and gingerol biosynthesis, starch and sucrose metabolism, sesquiterpenoid and triterpenoid biosynthesis, ribosome, photosynthesis-antenna proteins, phenylpropanoid biosynthesis, phenylalanine metabolism, pentose and glucuronate interconversions, PPAR signaling pathway, NOD-like receptor signaling pathway, metabolism of xenobiotics by cytochrome P450, linoleic acid metabolism, gap junction, flavonoid biosynthesis, estrogen signaling pathway, DNA replication, cysteine and methionine metabolism, cutin, suberine and wax biosynthesis, bile secretion, antigen processing and presentation, anthocyanin biosynthesis, and ABC transporters. Notably, a large number of genes related to flavonoid biosynthesis, anthocyanin biosynthesis, and phenylpropanoid biosynthesis were significantly differentially expressed in BXK compared to HXK (Figure 2B). Thus, we next analyzed the expression patterns of the DEGs in these pathways.

### 3.4. Analysis of DEGs Related to Anthocyanin Biosynthesis

As shown in Figure 3, the transcript levels of most genes involved in phenylpropanoid biosynthesis and flavonoid biosynthesis were significantly up-regulated in BXK-VS-HXK comparison. Interestingly, the expression levels of many potentially key genes for anthocyanin synthesis, including four potential chalcone synthase genes (CHS, g13809_i0, g12097_i0, g18851_i0, g36714_i0), one potential chalcone isomerase gene (CHI, g26337_i0), two potential flavonoid 3-hydroxylase gene (F3′H, g14710_i0 and g15005_i0), and one potential anthocyanidin synthase gene (ANS, g18981_i0), in flower of HXK were higher than that of BXK (Figure 3). These genes may be responsible for the synthesis of anthocyanidin, which colored the flower of HXK with red.

### 3.5. g8206_i0, a Potential UFGT Responsible for Glycosylating Anthocyanins

Anthocyanins can be further glycosylated by UFGT (uridine diphosphate-glucose:flavonoid 3-*O*-glucosyltransferase), leading to the formation of glycosylated anthocyanins. Significantly, glycosylated anthocyanins are the major anthocyanins that contribute to the color variation among different varieties. We found the abundance of 3-*O*-glucoside and 3-*O*-rutinoside glycosylated cyanidin, including delphinidin, malvidin, pelargonidin, and peonidin, are higher than other derivatives in HXK flowers (Table 1). Therefore, we next focused on glucosyltransferase, the key enzyme for the glycosylation of cyanidin, delphinidin, malvidin, pelargonidin, and peonidin.

KEGG analysis of DEGs between in BXK and HXK showed that two UFGTs were up-regulated (Figure 3). The expression of g8206_i0 in HXK, encoding a potential UFGT responsible for glycosylating cyanidin, delphinidin, and pelargonidin and producing cyanidin-3-*O*-glucoside, delphinidin-3-*O*-glucoside, and pelargonidin-3-*O*-glucoside, is more than 2.5 orders of magnitude more than that in BXK (Figure 4). Similarly, the expression of g5626_i0 in the flower of HXK is more than 4 orders of magnitude more than that in BXK (Figure 4). Moreover, the expression of g8206_i0 is prominent higher than g5626_i0 in flower of HXK. Thus, we reasoned that g8206_i0 is a key UFGT responsible for glycosylating cyanidin, delphinidin, and pelargonidin to produce cyanidin-3-*O*-glucoside, delphinidin-3-*O*-glucoside, and pelargonidin-3-*O*-glucoside, which play an important role in accumulating glycosylated anthocyanins in flowers of HXK.

### 3.6. Excavating Differentially Expressed Transcription Factors (TFs) in BXK-vs-HXK

The anthocyanin biosynthesis and modification are complex biological processes and strictly regulated by TFs, especially the MYB-bHLH-WD40 (MYB, basic helix-loop-helix (bHLH), WD40 proteins) complex [8,14]. Based on iTAK, PlnTFDB, and PlantTFDB, differentially expressed TFs were identified. As shown in Figure 5 and Appendix A, there were 73 differentially expressed TFs (40 up- and 33 down-regulated TFs) composed of SRF, MYB, MBD, Homeobox, CSD, HMG, zf-C2H2, E2F, HSF family TFs, and other TFs. Interestingly, 14 MYB family TFs (g25403_i0, g6939_i0, g21772_i0, g10657_i0, g27153_i0, g6484_i0, g20876_i0, g32204_i0, g29503_i0, g25712_i0, g30397_i0, g18595_i0, g18251_i0, g21878_i0) were up-regulated in BXK (Figure 5). Moreover, 10 MYB family TFs (g2206_i0, g33866_i0, g8716_i0, g5782_i0, g27919_i0, g19104_i0, g34207_i0, g30143_i0, g17765_i0, g36054_i0) were down-regulated in BXK (Figure 5).

## 4. Discussion

Flavonoids are further divided into anthocyanins, flavonoids, flavonols, and fla-vanones. Anthocyanins are anthocyanins that control the pink, red, blue, purple, and red-purple colors of flowers. Anthocyanins can be divided into three types: pelargonin, delphinidin, and cyanidin, which usually exist alone in different flowers. There are also three methoxyl substitution products of anthocyanins in plants, namely paeoniflorin, morning glory pigment, and mallow pigment. Geranium pigments are brick red, corn-flower pigments and peony pigments are purple-red, and delphinium pigments, morning glory pigments, and mallow pigments are blue-violet [32]. Flavonoids and flavonols are yellow or colorless; orangeone and chalcone often appear in the same petal together and appear dark yellow [33].

Cyclamen flowers’ color diversity are largely due to the differential accumulation of two categories of metabolites-carotenoids (yellow or deep orange pigments) and flavonoids (red, blue, or purple pigments). In general, the flower color can be affected by several factors. The first is pH as anthocyanins show different colors at different pH [34]. Second, anthocyanins can form metalloanthocyanin complex with Al^3+^ and Mg^2+^, which can give distinct flower colors [35]. Third, anthocyanins can also interact with other colorless molecules, such as copigments [35] or other anthocyanins [36], to enhance the color. Such a theory is also supported by the later intermolecular stacking theory [37]. For instance, polyacylated anthocyanins can form stable hydrophobic interactions that give blue color even without copigment or metal ions in a wide pH range. While these theories can explain the flower color to different extent, the molecular identifies underlying the color variation remain elusive. In this study, metabolic profiling of anthocyanin was performed to explore the dominating pigments endowing the cyclamen petal with red using white flowers as a control. In addition, we performed comparative transcriptome analysis to delineate key genes controlling the anthocyanin biosynthetic pathways.

Malvidin, peonidin and cyanidin derivatives are the predominant coloring pigment in cyclamen flowers. A total of seven anthocyanins, including peonidin 3-*O*-glucoside, cyanidin 3,5-di-*O*-glucoside, malvidin 3-*O*-glucoside, peonidin-rutinoside, peonidin 3,5-di-*O*-glucoside, malvidin 3,5-di-*O*-glucoside, and malvidin-rutinoside, were identified in flowers of ten cyclamen varieties [38]. Moreover, 108 anthocyanin metabolites were identified and quantified in cyclamen flower. Cyanidin-3-*O*-rutinoside, cyanidin-3-*O*-glucoside, delphinidin-3-*O*-glucoside, malvidin-3-*O*-glucoside, peonidin-3-*O*-rutinoside, quercetin-3-*O*-glucoside, and rutin were the main compounds in cyclamen flower. The content of these 3-*O*-glucoside or 3-*O*-rutinoside glycosylated cyanidin, delphinidin, malvidin, pelargonidin, and peonidin is prominent higher than the derivatives of them. Moreover, the content of most anthocyanin metabolites in flowers of “HXK” is higher than that in “BXK”.

The biosynthesis pathway of anthocyanins has been divided into two stages: anthocyanidin formation and anthocyanin modification such as glycosylation, methylation, and acylation [39]. Key enzymes for anthocyanidin biosynthesis include ANS, CHI, CHS, DFR, F3H, F3′H, and F3′5′H [6,8,9]. In our data, we found eight potentially key genes for anthocyanin synthesis, including four CHSs (g13809_i0, g12097_i0, g18851_i0, g36714_i0), one CHI (g26337_i0), two F3′Hs (g14710_i0 and g15005_i0), and one ANS (g18981_i0). Not surprisingly, our data showed that the expression of these eight genes are higher in the flower of HXK than that of BXK.

Anthocyanin modifications facilitate the formation of more anthocyanin subtypes. Glycosylation promotes the solubility, chemical stability, storage, and transportation of anthocyanins [40]. 3GT/UFGT (flavonoid 3-*O*-glucosyltransferase) is commonly considered as the first enzyme that catalyzes the glycosylation with several genes characterized in diverse plant species such as *Vitis vinifera*, *Petunia hybrida*, and *Freesia hybrida* [41,42]. Recently, an anthocyanin 5-*O*-glucosyltransferase (Cpur5GT) catalyzing the glycosylation of 3-glucoside-type anthocyanidins at the 5-*O*-position was identified from cyclamen [27]. Here, our transcriptomics data identified a potential UFGT (g8206_i0) that may execute the glycosylation of cyanidin (to give cyanidin-3-*O*-glucoside), delphinidin (to give delphinidin-3-*O*-glucoside), and pelargonidin (to give pelargonidin-3-*O*-glucoside). The mRNA of g8206_i0 in HXK flower is more than 2.5 orders more than that in BXK. The exact role of this gene needs further validation.

## Figures and Tables

**Figure 1 biology-11-01721-f001:**
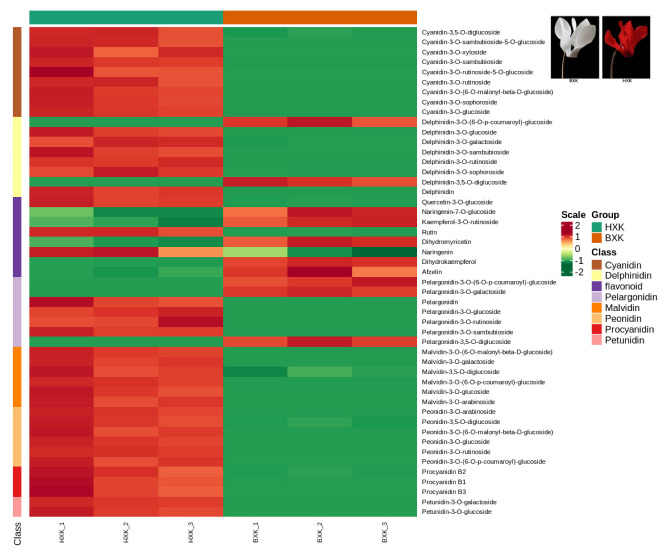
Hierarchical clustering based on anthocyanins in two cyclamen varieties with different flower colors.

**Figure 2 biology-11-01721-f002:**
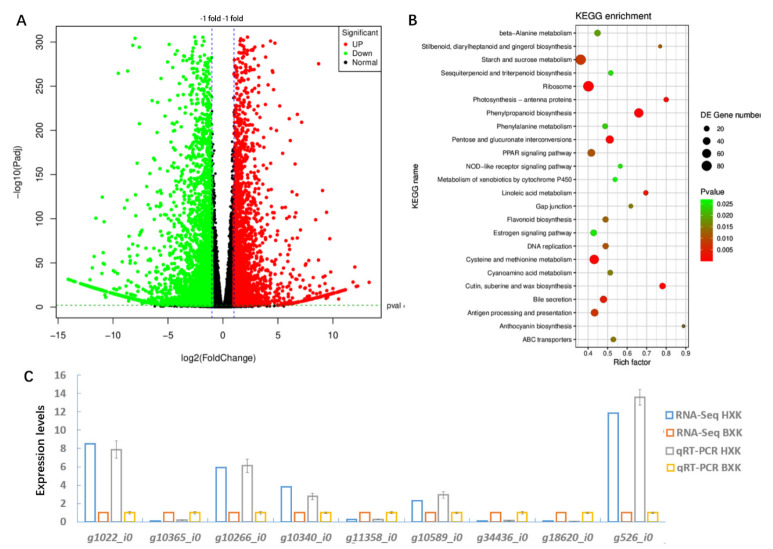
Transcriptional profiles of flowers of HXK and BXK. (**A**) Total numbers of DEGs. (**B**) KEGG pathway categories of DEGs. The X-axis (Rich factor) represents the proportion of DEG accounted for all genes of a specific KO term. The size of the point represents the number of DEGs. The Q value is the calibration of *p* value. (**C**) qRT-PCR analysis to verify the result of RNA-Seq. Randomly selected DEGs were labeled on the x axis. Relative expression levels between HXK and BXK (normalized to 1) for both the RNA-Seq and qRT-PCR data were shown. Means ± SE were shown from three replicates.

**Figure 3 biology-11-01721-f003:**
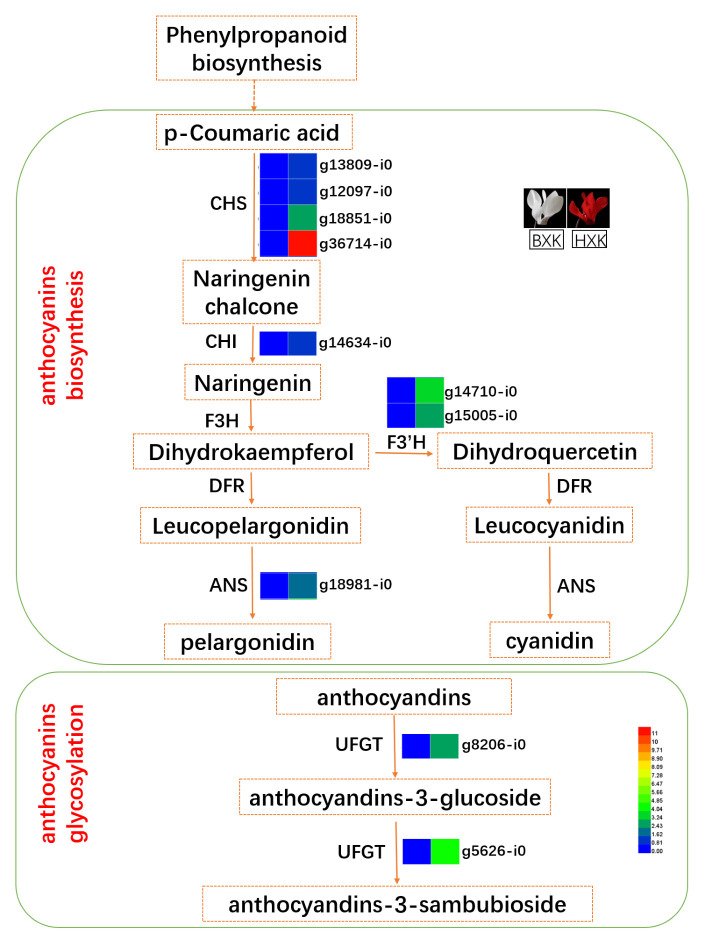
Putative key genes in anthocyanin biosynthesis and glycosylation. The expression heatmap represented log2 fold change of transcriptional level (BXK vs. HXK). ANS, anthocyanidin synthase; CHS, chalcone synthase; CHI, chalcone isomerase; DFR, dihydroflavonol 4-reductase; F3H, flavanone-3-hydroxylase; F3′H, flavonoid-3-hydroxylase; and UFGT, uridine diphosphate-glucose: flavonoid 3-O-glucosyltransferase.

**Figure 4 biology-11-01721-f004:**
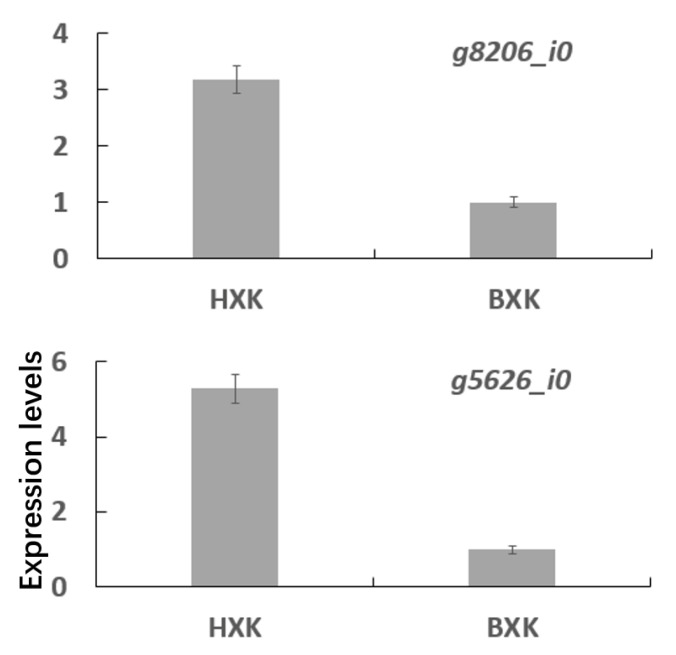
qRT-PCR analysis the expression of DEGs related to anthocyanin glycosylation. Means ± SE were shown from three biological replicates.

**Figure 5 biology-11-01721-f005:**
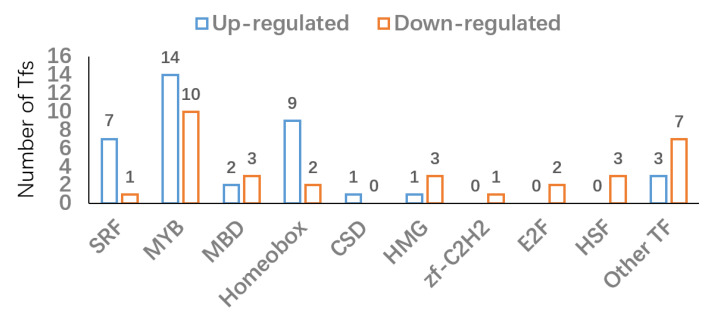
Number of up-regulated and down-regulated TFs.

**Table 1 biology-11-01721-t001:** The content of anthocyanins in flowers of two cyclamen varieties with different color (μg/g, dry weight).

Compounds	HXK_1	HXK_2	HXK_3	BXK_1	BXK_2	BXK_3
cyanidin-3,5-*O*-diglucoside	6.68	6.70	6.18	0.79	0.95	0.86
cyanidin-3-*O*-sambubioside-5-*O*-glucoside	0.21	0.21	0.19	0.00	0.00	0.00
cyanidin-3-*O*-xyloside	0.05	0.04	0.05	0.00	0.00	0.00
cyanidin-3-*O*-sambubioside	31.10	29.64	29.32	0.00	0.00	0.00
cyanidin-3-*O*-rutinoside-5-*O*-glucoside	0.59	0.49	0.49	0.00	0.00	0.00
cyanidin-3-*O-*rutinoside	188.05	188.79	169.85	0.00	0.00	0.00
cyanidin-3-*O*-(6-O-malonyl-beta-D-glucoside)	1.26	1.17	1.14	0.00	0.00	0.00
cyanidin-3-*O*-sophoroside	25.73	24.20	23.41	0.00	0.00	0.00
cyanidin-3-*O*-glucoside	117.54	112.72	109.88	0.00	0.00	0.00
delphinidin-3-*O*-(6-*O*-p-coumaroyl)-glucoside	0.00	0.00	0.00	0.04	0.04	0.04
delphinidin-3-*O-*glucoside	22.23	20.42	19.97	0.13	0.12	0.11
delphinidin-3-*O*-galactoside	0.23	0.25	0.24	0.02	0.03	0.03
delphinidin-3-*O*-sambubioside	0.10	0.09	0.09	0.00	0.00	0.00
delphinidin-3-*O*-rutinoside	0.13	0.13	0.13	0.00	0.00	0.00
delphinidin-3-*O*-sophoroside	0.23	0.26	0.24	0.03	0.03	0.03
delphinidin-3,5-*O*-diglucoside	0.00	0.00	0.00	0.07	0.06	0.06
delphinidin	0.72	0.67	0.68	0.00	0.00	0.00
malvidin-3-*O*-(6-*O*-malonyl-beta-*D*-glucoside)	53.95	50.78	49.96	0.17	0.17	0.17
malvidin-3-*O*-galactoside	68.52	63.64	65.12	0.00	0.00	0.00
malvidin-3-*O*-(6-*O*-p-coumaroyl)-glucoside	0.31	0.30	0.29	0.00	0.00	0.00
malvidin-3-*O*-glucoside	2969.14	2746.50	2602.56	0.00	0.00	0.00
malvidin-3-*O*-arabinoside	0.33	0.29	0.31	0.00	0.00	0.00
pelargonidin-3-*O*-(6-*O*-p-coumaroyl)-glucoside	0.00	0.00	0.00	0.09	0.09	0.10
pelargonidin-3-O-galactoside	0.00	0.00	0.00	0.07	0.07	0.07
pelargonidin	0.07	0.06	0.06	0.00	0.00	0.00
pelargonidin-3-*O*-glucoside	0.06	0.06	0.07	0.00	0.00	0.00
pelargonidin-3*-O-*rutinoside	0.46	0.47	0.54	0.00	0.00	0.00
pelargonidin-3*-O*-sambubioside	3.49	3.26	3.27	0.00	0.00	0.00
pelargonidin-3,5-*O*-diglucoside	0.02	0.02	0.02	0.57	0.63	0.58
peonidin-3-*O*-arabinoside	0.05	0.04	0.04	0.00	0.00	0.00
peonidin-3,5-*O*-diglucoside	12.96	12.31	11.93	3.27	3.52	3.18
peonidin-3-*O*-(6-*O-*malonyl-beta-*D*-glucoside)	0.70	0.61	0.64	0.00	0.00	0.00
peonidin-3-*O*-glucoside	22.29	21.18	20.67	0.02	0.02	0.01
peonidin-3-*O*-rutinoside	742.94	733.11	716.02	0.00	0.00	0.00
peonidin-3-*O*-(6-*O*-p-coumaroyl)-glucoside	14.85	13.20	13.96	0.00	0.00	0.00
petunidin-3-*O-*galactoside	17.94	17.30	17.09	0.00	0.00	0.00
petunidin-3-*O*-glucoside	3.83	3.55	3.45	0.00	0.00	0.00
procyanidin B2	7.00	6.55	5.93	0.41	0.52	0.43
procyanidin B1	0.24	0.21	0.21	0.01	0.01	0.01
procyanidin B3	0.48	0.42	0.39	0.00	0.00	0.00
quercetin-3-*O*-glucoside	380.52	353.15	356.23	37.09	40.72	38.06
kaempferol-3-*O*-rutinoside	7.57	7.12	6.35	14.08	14.88	15.02
rutin	694.99	696.19	640.73	45.35	50.33	45.72
dihydromyricetin	3.64	3.22	3.01	8.28	9.09	8.85

**Table 2 biology-11-01721-t002:** Number of transcripts, unigenes, and conden genes with different lengths after splicing.

Transcript Length Interval	200–500 bp	500–1k bp	1k–2k bp	>2k bp	Total
Number of transcripts	62,606	25,307	31,355	29,654	148,922
Number of unigenes	62,588	21,048	14,300	13,006	110,942
Number of conden genes	302	3345	7946	10,029	21,622

**Table 3 biology-11-01721-t003:** Total reads number, clean reads number, Q20 value, Q30 value, total mapped reads, and unique mapped reads based on the RNA-Seq data in libraries of HXK and BXK. _1, _2, and_3 represent the three biological replicates of each sample. Q20 and Q30 represent the percentage of bases with a Phred value greater than 20 and 30, respectively.

Samples	Total_Reads	Clean_Reads	%>Q20	%>Q30	Mapped Reads (%)	Secondary Alignments (%)	Unique Mapped (%)
BXK_1	81,727,748	81,727,706	97.81%	93.52%	67,113,867 (67.24)	18,086,546 (18.12)	49,027,321 (49.12)
BXK_2	93,641,990	93,641,948	97.87%	93.71%	76,691,067 (67.22)	20,445,351 (17.92)	56,245,716 (49.3)
BXK_3	106,129,474	106,129,422	97.96%	93.97%	86,499,777 (66.63)	23,691,204 (18.25)	62,808,573 (48.38)
HXK_1	91,740,416	91,740,386	97.92%	93.85%	77,487,325 (68.14)	21,970,557 (19.32)	55,516,768 (48.82)
HXK_2	96,117,694	96,117,654	97.73%	93.31%	81,692,006 (68.9)	22,455,670 (18.94)	59,236,336 (49.96)
HXK_3	94,944,072	94,944,020	97.78%	93.44%	81,438,637 (69.49)	22,253,903 (18.99)	59,184,734 (50.5)

## Data Availability

Data from this study are available from the corresponding author upon reasonable request.

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
