# Peer review of "Anthocyanins Profiling Analysis and RNA-Seq Revealed the Dominating Pigments and Coloring Mechanism in Cyclamen Flowers"

_biology, 2022, doi:10.3390/biology11121721_

Round 1

Reviewer 1 Report

Anthocyanins profiling and RNA-Seq were applied to explore dominating pigments and coloring mechanism in flower of two cyclamen genotypes with different colors. The whole paper is easy to understand, but some questions should be solved before it is accepted. The grammar should be checked carefully through the paper. 

Could you please explain the reason why anthocyanin shows red in the flower? Why not purple or blue?

Except for anthocyanin, does any other factor cause red color in flower?

Could you please add a HPLC chromatograph of anthocyanins in two cyclamen genotypes. 

Line 68, can anthocyanin show brown color? In which pH? 

Line 87 (1-5) should be changed to [1-5]

Line 90 “different color” should be different colors

Line 133-134 so many (), please check

Line 142, what does “40” means?

Each figure should be cited in the text. You did not cite figure 1A.

Line 203, “suggest” should be suggesting

Line 214-216 reword

Line 360-361 reword

Author Response

Dear Reviewers:

Thanks for your suggestions. Your suggestions are valuable for improving our manuscript. We believe we have solved all your comments in the new version.

Please find it in the attachment。

Kind regards,

Reviewer 2 Report

The manuscript “Anthocyanins profiling analysis and RNA-Seq revealed the dominating pigments and coloring mechanism in cyclamen flowers” analyzed the differential anthocyanin components and expression levels of related genes for revealing mechanism of different floral colors of cyclamen flowers. However, the following comments needs to be addressed before publishing.

1. Please provide LC-MS/MS chromatograms corresponding to Table 1. 2. Line 84, normally it should be ‘bHLH’ instead of ‘Bhlh’. 3. The image of Figure 1 is not clear. Please provide a high-resolution version. 4. There are some grammar problems in the manuscript. Such as Line 252-256. The whole manuscript should be checked again.

5. Anthocyanins like 3-O-glucoside or 3-O-rutinoside glycosylated cyanidin, delphinidin, malvidin, pelargonidin, peonidin, and flavonols such as rutin were detected in cyclamen flowers, but Figure 3 only shows anthocyanin biosynthesis pathway to pelargonidin and cyanidin. Please modify Figure 3 according to detected anthocyanins.

6. Lines 207 and 266: you mentioned Figure 3A and Figure 3B, but there are no image A and B in Figure 3.

7. How did you make sure that unigenes g8206_i0 and g5626_i0 are responsible for 3-O-glucoside and 3-O-sambubioside of anthocyanins, respectively?

8. Description for differential expressed regulatory genes encoding transcription factors, such as MYBs and bHLHs were not enough in this manuscript. For example, which subfamilies these MYBs and bHLHs belong to? Among them, which could be candidates that activate or repress expression of anthocyanin biosynthetic genes?

9. Normally, anthocyanins are synthesized in petals at early flowering stages. So how about the expression profiles of anthocyanin-related genes and TFs during whole flowering process of two cyclamen flowers?

Author Response

Dear reviewer:
Thanks for your comments. Your suggestions are valuable in improving our manuscript. We believe we have addressed all your concerns in the new version.
Please find it in the attachment.
Kind regards
Guiling Liu

Reviewer 3 Report

Authors have described about differential accumulation of different anthocyanins and its related genes in two stages of Cyclamen purpurascen. Following things need to be looked into:

Section: 2.2: Authors have not mentioned number of replicates as part of RNA sequencing experiment.

2.6.4: How was the quantification of anthocyanin compounds done? Authors need to elaborate on this. Were standards used for all 211 or so compounds?

Why only glycosyl transferase for anthocyanin glycosylation was intensively studied including qRTPCR?

Table 2 Number of coden genes? The word 'coden' seems to be in appropriate 

Results and Discussion section should have more information regarding KEGG enrichment analysis fold change etc.

The Gi numbers are not visible in Figure 3. Kindly revise

Author Response

(The authors gave the same response as above.)

Round 2

Reviewer 1 Report

Line 72-73, the font size is not correct.

The supplementary pictures are very blurry, please change. 

Author Response

尊敬的审稿人:
感谢您的评论。您的建议对改进我们的手稿很有价值。我们相信我们已经在新版本中解决了您的所有问题。

亲切的问候
刘桂玲

Reviewer 2 Report

The revised manuscript has been more improved according to the previous suggestions. Therefore, I feel this study is appropriate for publication in Biology.

Author Response

Dear reviewer:
Thanks for your comments. 

Kind regards
Guiling Liu